# Enrichment of Rice Flour with Almond Bagasse Powder: The Impact on the Physicochemical and Functional Properties of Gluten-Free Bread

**DOI:** 10.3390/foods14132382

**Published:** 2025-07-05

**Authors:** Stevens Duarte, Janaina Sánchez-García, Joanna Harasym, Noelia Betoret

**Affiliations:** 1Instituto Universitario de Ingeniería de Alimentos—FoodUPV, Universitat Politècnica de València, Camino de Vera s/n, 46022 Valencia, Spain; steduase@doctor.upv.es (S.D.); jmsanga1@upv.es (J.S.-G.); 2Department of Biotechnology and Food Analysis, Wroclaw University of Economics and Business, 53-345 Wroclaw, Poland; joanna.harasym@ue.wroc.pl

**Keywords:** food waste, almond bagasse powder, hot air-drying, lyophilised, bread, flour, storage, techno-functional properties, antioxidant properties, rheological properties

## Abstract

Almond bagasse, a by-product of almond milk production, is rich in fibre, protein, polyunsaturated fatty acids, and bioactive compounds. Its incorporation into food products provides a sustainable approach to reducing food waste while improving nutritional quality. This study explored the impact of enriching rice flour with almond bagasse powders—either hot air-dried (HAD60) or lyophilised (LYO)—at substitution levels of 5%, 10%, 15%, 20%, 25%, and 30% (*w*/*w*), to assess effects on gluten-free bread quality. The resulting flour blends were analysed for their physicochemical, techno-functional, rheological, and antioxidant properties. Gluten-free breads were then prepared using these blends and evaluated fresh and after seven days of refrigerated storage. The addition of almond bagasse powders reduced moisture and water absorption capacities, while also darkening the bread colour, particularly in HAD60, due to browning from thermal drying. The LYO powder led to softer bread by disrupting the starch structure more than HAD60. All breads hardened after storage due to starch retrogradation. The incorporation of almond bagasse powder reduced the pasting behaviour—particularly at substitution levels of ≥ 25%—as well as the viscoelastic moduli of the flour blends, due to fibre competing for water and thereby limiting starch gelatinisation. Antioxidant capacity was significantly enhanced in HAD60 breads, particularly in the crust and at higher substitution levels, due to Maillard reactions. Furthermore, antioxidant degradation over time was less pronounced in formulations with higher substitution levels, with HAD60 proving more stable than LYO. Overall, almond bagasse powder improves the antioxidant profile and shelf-life of gluten-free bread, highlighting its value as a functional and sustainable ingredient.

## 1. Introduction

The valorisation of plant-based by-products has emerged as a pivotal strategy in the agri-food sector, aiming to enhance sustainability and reduce environmental impact [1]. By-products from the fruit and vegetable processing industry—such as hulls, shells, and skins—represent a valuable source of bioactive compounds, including fibre, antioxidants, and various phytochemicals [2,3]. Traditionally underutilised, these by-products are now recognised for their rich composition of bioactive compounds, such as polyphenols and unsaturated fatty acids, offering potential applications in food, cosmetic, and pharmaceutical industries [4,5]. Almond processing generates substantial residues that constitute approximately 70–85% of the total fruit mass [4,5]. Among these residues, almond bagasse has garnered attention for its nutritional value and functional properties [6]. It is a by-product whose generation is associated with the production of the almond vegetable drink, and consequently, its increase is associated with the increase in its production. It is worth noting that the almond milk market was estimated at USD 5.49 billion in 2024 and is expected to reach USD 9.61 billion by 2029, growing at a CAGR of 11.85% over the period from 2024 to 2029 [7]. When converted into powder, this by-product retains important nutritional and functional properties, making it a promising ingredient for the enrichment of food products and the enhancement of their technological performance. Beyond reducing waste, this practice fosters the use of innovative, sustainable, and health-promoting components by the food industry [2,8]. This approach aligns with the principles of a circular economy, promoting the efficient use of resources and the development of value-added products [9].

The enrichment of traditional flours such as wheat, maize, and rice with powders derived from by-products in bread production offers multiple advantages. Firstly, these by-products are rich in dietary fibre and bioactive compounds, which can enhance the nutritional profile of bread. For instance, incorporating apple and bottle gourd pomace into wheat flour has been shown to increase fibre content and total phenolic compounds, thereby improving the bread’s antioxidant capacity [10]. Additionally, these enrichments can improve the technological properties of bread, such as moisture retention and shelf-life, due to the presence of hydrophilic compounds in the by-products [11]. However, it is important to consider the potential impact on sensory attributes; studies have found that incorporating up to 5% of these by-products maintains acceptable taste and texture in bread [11]. Overall, the strategic replacement of conventional flours with by-products not only enhances the nutritional and functional qualities of bread but also supports environmental sustainability.

Among the promising by-products for such applications, almond bagasse stands out due to its favourable nutritional composition and technological functionality. Processing almond bagasse into a powdered form significantly enhances its properties, such as emulsifying activity and stability, thereby positioning it as a high-value functional ingredient for the food industry [12]. These enhancements not only provide technological advantages but also contribute to improving the sensory and nutritional quality of baked goods, supporting the development of healthier and more sustainable alternatives within the bakery sector. Its incorporation has been shown to improve texture, extend product shelf life, and reduce the use of less healthy fats [13]. In a recent study, Duarte, Harasym et al. [13] investigated the use of almond bagasse powder as a partial substitute for wheat flour in biscuit formulations. The results demonstrated that wheat flour exhibited no emulsifying activity, while the bagasse powder reached 20% when hot-air dried and 59% when freeze-dried. This functionality is attributed to its content of proteins and beneficial lipids, as well as to structural changes induced by the drying process. Incorporating this ingredient into gluten-free baked products would improve nutritional properties while also promoting more sustainable practices in the food industry. The benefits of valorising the by-product and improving the nutritional value of gluten-free bread would increase when the level of substitution increases. Furthermore, almonds are a nut whose sensory properties are valued by consumers, suggesting that the changes in sensory properties associated with a high level of substitution could be favourable. However, incorporating this ingredient could also lead to undesirable characteristics. Its high fat and fibre content, together with the formation of compounds during air drying, could impart residual and rancid flavours to the final product. Furthermore, it requires additional treatments that increase production costs.

In this context, the present study aims to evaluate the effect of partially replacing rice flour with almond bagasse powder—processed via freeze-drying or hot-air drying—on the physicochemical, techno-functional, rheological, antioxidant, and textural properties of gluten-free breads, in order to assess its potential as a functional ingredient in gluten-free bakery formulations.

## 2. Materials and Methods

### 2.1. Almond Bagasse and Almond Bagasse Powder Production

Almonds (*Prunus dulcis* var. *dulcis*) purchased from a local supermarket were hydrated in tap water at a 1:2 (*w*:*w*) ratio for approximately 12 h, resulting in a weight increase of around 40%. The rehydrated almonds were subsequently blended with water at a 1:9 (*w*:*w*) ratio and processed into a fine mixture using a domestic food processor (Thermomix^®^, Vorwerk, Madrid, Spain) operating at 10,000 rpm for 20 s. The resulting grinding was sieved under atmospheric pressure through a stainless steel mesh with a 500 µm aperture. The almond bagasse retained on the sieve was collected for subsequent analysis and processing, accounting for approximately 82% of the weight of the hydrated almonds.

For the production of dehydrated almond bagasse, the collected material was evenly spread into a thin layer (5–7 mm thick) on plastic grids featuring a nominal 2 mm opening. This was then subjected to hot air drying in a convective dryer (Pol-Eko Aparatura, Katowice, Poland) using a cross-flow of air at 10 m/s and a temperature of 60 °C for 10 h, until the water activity (aw) dropped below 0.3, resulting in the air-dried almond bagasse (HAD60). Additionally, a freeze dryer (Telstar, Barcelona, España) was used to obtain lyophilised almond bagasse (LYO) from material previously frozen at −40 °C for 24 h, followed by sublimation at −45 °C (condenser temperature) and 0.1 mbar for 48 h.

Subsequently, both types of dehydrated almond bagasse were ground using a domestic food processor (Thermomix^®^, Vorwerk, Spain), first at 4000 rpm for 20 s with 5 s pauses, and then at 10,000 rpm for a further 20 s, also with 5 s pauses, thus obtaining almond bagasse powders. The powders were then stored at 20 °C in opaque glass jars to prevent deterioration and oxidative reactions. Finally, the almond bagasse powders (HAD60 and LYO) were incorporated into rice flour at substitution levels of 5, 10, 15, 20, 25, and 30%, and the resulting blends were subsequently characterised.

### 2.2. Bread Production Using Almond Bagasse Powder

The gluten-free bread was prepared following the protocol described by Marti et al. [14]. Bread was made using only rice flour, which was considered the control. This formulation consisted of 100 g of rice flour, 1.5 g of salt, 1 g of dry bakery yeast, and water corresponding to 44% hydration. Subsequently, breads containing almond bagasse powder, either air-dried or lyophilised, were prepared as partial substitutes for the rice flour. In these formulations, a constant water content of 44% hydration was maintained across all treatments (consistent with the control formulation), with 100 g of flour mixture as the base, and substitution levels of 5, 10, 15, 20, 25, and 30% (*w*/*w*). This approach was deliberately chosen to evaluate the specific impact of almond bagasse substitution on bread matrix properties under standardised hydration conditions, rather than optimising water content for each individual blend. The flours, along with the other ingredients, were kneaded for 5 min using a 1000 W mixer at speed 5 (MUM58231, BOSCH, Stuttgart, Germany). The dough was then divided into 100 g portions, placed in moulds measuring 75 mm × 75 mm × 75 mm, and fermented at 30 ± 1 °C for 180 min in a 2 kW fermentation chamber (823HO, Bartscher, Salzkotten, Germany). Finally, the samples were baked at 200 ± 1 °C for 22 min in a convection-steam oven (Nano, Grafen, Robakowo, Poland). After baking, the bread was allowed to cool at room temperature for analysis as fresh bread, and then stored in refrigeration at 4 °C for 7 days for subsequent analysis.

### 2.3. Analytical Determinations

#### 2.3.1. Physicochemical Properties

Moisture content

The moisture content of the rice flour and blended flours was determined using the Official Method AACC44-19 Moisture-Air-Oven Method [15], which involves drying at 135 °C.

Colour

Colour analysis was performed on rice flour and blended flours, as well as on the crumb and crust of freshly baked bread and bread stored for seven days, using a Konica Minolta CR-310 chromameter (New York, NJ, USA), considering a standard illuminator D65 and a standard observer of 2°. The colour parameters were determined as CIE-L*a*b* coordinates. The colour differences (ΔE) between rice flour and blended flours, and between freshly baked and stored breads were calculated according to Equation (1):(1)$$∆E=\sqrt{{\left(∆L^{*}\right)}^{2}+{\left(∆a^{*}\right)}^{2}+{\left(∆b^{*}\right)}^{2}}$$

Relative weight and specific volume

The weight of the baked breads was determined using an Axis AD1000 balance (Axis, Gdansk, Poland), and the volume was measured using a 3D Scanner V2 analyser with Quickscan (MatterandForm, Northamptonshire, UK). The relative weight was calculated as the ratio of the weight of bread prepared with the flour blend compared to the weight of bread prepared with rice flour, fresh or stored, as appropriate.

Water activity

Water activity of the baked breads was measured with an AquaLab 3TE analyser (Decagon Devices, Inc., Pullman, WA, USA) at 25 °C.

Reducing sugars

The determination of reducing sugars of breads was carried out by the dinitrosalicylic acid (DNS) method following the protocol described by Hu et al. [16] and Başkan et al. [17]. A 500 µL aliquot of the extract was taken and mixed with 250 µL of DNS reagent (1 g 3,5- DNS in 100 mL of 4 M NaOH solution; both reagents were obtained from Sigma-Aldrich, St. Louis, MO, USA). The sample was heated in a boiling water bath for 5 min, and then allowed to cool to 50–60 °C. The sample was diluted with 3 mL of distilled water, and the absorbance was measured at 530 nm (UV–VisUltrospec 2000, Pharmacia Biotech, Piscataway, NJ, USA). A glucose standard curve was performed, and the results were expressed as mg glucose/g_dm_.

#### 2.3.2. Techno-Functional Properties

Water Absorption Capacity (WAC) and Oil Absorption Capacity (OAC)

The Water Absorption Capacity (WAC) and the Oil Absorption Capacity (OAC) of the rice flour and blended flours were determined according to the methodology described by Abebe et al. [18] with some modifications. An amount of 1 g of flour was weighed into a centrifuge tube and mixed with 10 mL of distilled water or vegetable oil. The contents were vigorously vortexed at high speed (Heidolph Reax, Schwabach, Germany) for 30 s until a homogeneous mixture was obtained and allowed to stand for 10 min at room temperature. After that, the samples were centrifuged at 3000× *g* (Thermo Fisher Scientific, Waltham, MA, USA) and the supernatant was collected and weighed. The results were expressed as g of water or oil retained/g_flour_.

#### 2.3.3. Rheological Properties

Pasting

The viscometric profile of the rice flour and blended flours was obtained following the procedure of Harasym et al. [19], in accordance with the ICC 162 Standard Method [20], using a Rapid Viscosity Analyser (RVA-4500, Perkin Elmer, Waltham, MA, USA). A quantity of 3.5 g of flour was taken and placed in an RVA container, and distilled water (as solvent) was added until a total weight of 28.5 g was achieved. Flour suspension was held at 50 °C for 1 min, and then the temperature was raised to 95 °C at a rate of 5 °C/min and held at this temperature for 5 min. It was then cooled to 50 °C at a rate of 5 °C/min and maintained at this temperature for the last 4 min. The stirring speed was set at 960 rpm for the first 10 s and then reduced to 160 rpm for the remaining analysis. The samples were analysed in triplicate. The results for the following parameters were obtained using the TCW3 software v.1.0 (Perkin Elmer, Beaconsfield, Inglaterra. UK): peak viscosity (PV), trough viscosity (TV), breakdown (BD = PV − TV), final viscosity (FV), and setback (ST = FV − TV), as well as the pasting temperature and peak time.

Rheological analysis

Dynamic oscillatory tests were conducted on the starch paste obtained after the pasting stage, where rice flour and flour blends were hydrated, heated to 95 °C, and then cooled to 50 °C. These tests were performed using an Anton Paar MC102 rheometer (Anton Paar, Stuttgart, Germany). A parallel plate geometry with a 40 mm diameter and serrated steel surface was used, maintaining a 1 mm working space, with the temperature precisely controlled at 25 °C by a KNX2002 thermal controller. Prior to testing, the starch paste was placed on the plate, excess material carefully removed, and the sample allowed to equilibrate for 5 min. The viscoelastic properties were assessed by measuring the storage or elastic modulus (G′) and loss or viscous modulus (G″) through a frequency sweep from 10 to 1 Hz within the linear viscoelastic region, applying a constant stress of 1 Pa. Each sample was performed in triplicate.

Texture

Texture analysis was carried out on the gels obtained after the pasting stage, during which rice flour and flour blends were hydrated, heated to 95 °C, and then cooled to 50 °C. The resulting starch paste was moulded into cylinders of 2 cm diameter and 2 cm height, and then stored under refrigeration to achieve firm gels. Additionally, the texture of the baked breads prepared with rice flour and the flour blends was evaluated by cutting crumb samples into 2 cm cubes. Both types of samples (gels and breads) were analysed in triplicate on a texture analyser (AXIS, Gdansk, Poland), using a double compression test (Texture Profile Analysis, TPA) with a cylindrical aluminium probe of 20 mm diameter. Measurements were taken under 50% sample deformation, at a speed of 1 mm/s, with a 30 s interval between the first and second compressions. Maximum hardness (N) during the first and second compressions was determined using the AXIS FM v.2_18 (Axis, Gdansk, Poland) software. It should be noted that for highly brittle samples that fracture during the first compression, second compression-derived parameters may be less reliable than those obtained from intact samples.

#### 2.3.4. Antioxidant Properties

The antiradical capacity and total phenol content were measured for different flours (rice flour and blended flours), fresh bread (crumb and crust), and 7-day stored bread (crumb and crust). Initially, sample extracts were prepared by double extraction by weighing 0.5 g of sample and mixing with 8 mL of water. The samples were then shaken in a laboratory orbital shaker (WU4, Premed, Marki, Poland) at room temperature for 2 h. Subsequently, the samples were centrifuged at 3500× *g* for 10 min (MPW-350, MPW, Warszawa, Poland). The supernatants were collected and pooled to be used for the following determinations.

Antiradical capacity

The antiradical capacity of the samples was determined by DPPH (2,2-diphenyl-1-picrylhydrazyl) and FRAP (Ferric reducing antioxidant power) methods according to the methodology described by Brand-Williams et al. [21], Nixdorf & Hermosín-Gutiérrez [22], and Ol˛ Edzki et al. [23], respectively.

For the DPPH assay, a 0.1 mM working solution of DPPH in pure methanol was prepared until an absorbance of 0.9 ± 0.1 at 517 nm was obtained. Then, a 34.5 µL aliquot of the extract was taken and reacted with 1 mL of the DPPH working solution for 20 min in the dark and the absorbance was measured at 517 nm. For the FRAP assay, the following stock solutions were prepared: (A) 300 µM of acetate buffer (pH 3.6), (B) 10 µM of TPTZ (2,4,6-tripyridyl-s-triazine) in 40 µM HCl, and (C) 20 µM of FeCl_3._ Then, a fresh working solution was prepared by mixing solutions A, B, and C in a 10:1:1 ratio, respectively. A 35.5 µL aliquot of the extract was taken and mixed with 998 µL of the working solution in a test tube. They were incubated at 36–38 °C for 15 min in the dark, and the absorbance was measured at 593 nm. The results of antioxidant activity by DPPH and FRAP methods were expressed as µmol trolox/g_dm_, using a standard trolox curve.

Total phenolic content (TPC)

The determination of the total phenol content was carried out using the Folin–Ciocalteu method described by Harasym et al. [19]. A 20 µL aliquot of the extract was taken and mixed with 1.58 mL of distilled water, followed by 100 µL of Folin–Ciocalteu reagent. The mixture was allowed to react for 5–8 min at room temperature. Then, 300 µL of a saturated sodium carbonate (Na_2_CO_3_) solution was added and incubated at 38 °C for 30 min (MLL147, AJL Electronics, Kraków, Poland) in the dark. The absorbance was measured at 765 nm in a UV/Vis spectrophotometer (UV–VisUltrospec 2000, Pharmacia Biotech, Piscataway, NJ, USA). A calibration curve was performed using gallic acid as a standard, and the results were expressed as mg GAE/g_dm_.

### 2.4. Statistical Analysis

The samples were analysed in triplicate, and the results are presented as mean values ± standard deviation. Statistical analysis was conducted using Statgraphics Centurion-XV software. Analysis of variance (one-way ANOVA and multifactorial ANOVA) was performed at a 95% confidence level (*p* ≤ 0.05) to assess the effect of rice flour substitution with almond bagasse powder. This statistical evaluation allowed the identification of significant differences among the formulations and provided insight into the influence of substitution levels on the measured properties.

## 3. Results

### 3.1. Physicochemical and Techno-Functional Properties of Flour and Bread Under Fresh and Stored Conditions

The incorporation of almond bagasse powder into rice flour not only modifies the chemical composition but also alters the techno-functional and optical properties of the resulting flour blends. Table 1 presents the moisture content (Xw), Water Absorption Capacity (WAC), Oil Absorption Capacity (OAC), and colour parameters of rice flour (control), almond bagasse powder dehydrated by hot air at 60 °C (HAD60) and by lyophilisation (LYO), as well as of the flour blends formulated by partially enriching rice flour with almond bagasse powder, which was incorporated at concentrations ranging from 5% to 30%. Significant differences (*p* ≤ 0.05) in moisture content were observed between rice flour and flour blends with almond bagasse powder. As the substitution level increases, the moisture content decreases. This reduction can be attributed to the low moisture content of both lyophilised and hot air-dried almond bagasse powder, with values close to 1–2% [6], which contributes to a decrease in moisture values when combined with rice flour.

Regarding WAC, significant differences were observed depending on the substitution level. It can be noted that as the substitution percentage increases, the WAC tends to decrease. This could be due to the high fat content in almond bagasse [6], which could hinder proper interaction with water molecules [24]. With regard to the OAC, the differences were significantly higher when lyophilised almond powder was added at a percentage between 5% and 15%. Samples containing hot air-dried or lyophilised powder at percentages between 20% and 30% exhibited OAC values comparable to the control sample. This tendency could be attributed to the increased fibre content introduced by the almond bagasse powder, which would result in greater competition for water and oil absorption [25]. In addition, other factors that can influence the functional properties of powders include pH, the presence of hydrophilic components such as fibre and free hydroxyl groups, as well as flour porosity and particle size [26]. With regard to porosity, it can be asserted that lyophilised powder is more porous, and lower substitution percentages result in less compaction and caking of the mixture, allowing it to interact more easily with water and oil molecules. Similar findings have been reported when replacing rice flour with amaranth or cabbage flour [25,27].

With respect to colour evaluation, all samples exhibited significant differences in the L*, a*, b*, C*, and ΔE parameters compared to the rice flour (control). The addition of these powders resulted in samples with yellow–reddish tones, attributable to browning and oxidation processes. According to Bodart et al. [28], when the ΔE value is below 1, colour differences are not perceptible to the human eye. If ΔE is between 1 and 3, variations are subtle but detectable. However, when ΔE exceeds a value of 3, colour differences become clearly noticeable. Except for the LYO-5% sample, all other samples exhibited ΔE values greater than 3, indicating clearly perceptible colour changes. Nevertheless, the LYO samples presented less pronounced colour variations compared to the HAD60 samples, probably because the lyophilisation process is carried out at low temperatures and under vacuum, which minimises oxidation.

Key technological parameters were assessed to characterise the quality of gluten-free bread, both in its fresh form and after 7 days of refrigerated storage. The analysis focused on the impact of almond bagasse powder type (LYO or HAD60) and substitution level on the bread’s physical properties. Table 2 presents the results for water activity, relative weight, specific volume, and colour properties of both fresh and 7-day stored bread. These breads were made with rice flour (control) and with flour blends where rice flour was partially substituted by almond bagasse powder, either HAD60 or LYO, with substitution levels ranging from 5% to 30%. Regarding water activity, no significant differences were observed between fresh and 7-day stored bread, with values ranging between 0.984 and 0.994. According to Chareonthaikij et al. [29], a higher water content can improve bread texture, which is reflected in an increase in specific volume. Concerning the latter parameter (specific volume), the obtained results showed a differentiated behaviour depending on the type of almond bagasse powder used as a rice flour substitute. In formulations prepared with hot air-dried powder, the specific volume decreased progressively as the substitution level increased (5–30%). This trend could be attributed to the higher density and lower porosity of this type of powder, which would limit gas retention during fermentation and baking [30]. Furthermore, its high Water Absorption Capacity could reduce the availability of moisture for bread structure development, resulting in a denser and more compact crumb [6]. In contrast, formulations containing lyophilised almond bagasse powder showed an increase in specific volume with increasing substitution levels. This behaviour suggests that the more porous and lightweight structure of the lyophilised powder enhances dough aeration, facilitating expansion during fermentation and baking [31]. In addition, its ability to retain water in a more balanced way could contribute to greater stability of the starch matrix and other rice flour components, allowing a fluffier crumb and improved bread development. After 7 days of refrigerated storage, the initial trend persisted; breads made with lyophilised powder exhibited the highest specific volume values. This behaviour could be explained by internal moisture redistribution, which softens the crumb and promotes slight expansion or volume preservation [32]. Likewise, the porous structure of the lyophilised powder could provide greater stability and gas-holding capacity, preventing crumb collapse during storage [33]. Lastly, a potential loss of mass (water) without volume reduction could also contribute to the increase in specific volume [34].

With respect to relative weight, neither fresh nor 7-day stored bread showed statistically significant differences between each other. In fresh breads, the relative weight was equal to 1, indicating that partial substitution of rice flour by almond bagasse powder (5–30%, whether hot air-dried or lyophilised) did not significantly affect the weight compared to the control bread. Conversely, breads stored under refrigeration for 7 days exhibited relative weight values greater than 1, suggesting that the substituted breads weighed more than the control and were therefore different. This change could be attributed to a higher moisture loss in the control bread during storage, while the formulations with almond bagasse powder showed better water retention. This may be related to the moisture-holding capacity of the substitute ingredients, which reduces dehydration during storage. Similar results have been reported in breads made with almond skin and potato flour, where higher water retention was observed compared to control breads [35,36].

The colour parameters L*, a*, b*, and C* and colour difference (ΔE) were also evaluated in both fresh and 7-day stored breads. A progressive decrease in L* values was observed with increasing levels of almond bagasse powder substitution, indicating bread darkening. This effect was significantly more pronounced in the samples containing hot air-dried powder, likely due to browning reactions occurring during thermal drying [37]. The a* and b* values also increased across all formulations as the substitution level rose, indicating a shift towards more reddish and yellowish tones. This behaviour is attributed both to the natural pigments present in almond bagasse and to thermal reactions during the drying process. In particular, breads containing lyophilised powder exhibited the highest b* values, reflecting a more yellow colour, possibly due to better preservation of bioactive colour-related compounds [33,38]. Regarding the colour difference (ΔE), the values recorded for the HAD60-30% and LYO-30% samples were close to 4, indicating colour differences clearly perceptible to the human eye. After 7 days of refrigerated storage, the observed trends in L*, a*, and b* values were maintained when compared to fresh breads. However, a slight increase in L* and b* values, together with a decrease in a*, was recorded, which could be attributed to several factors. Firstly, the redistribution of moisture from the crumb to the crust during refrigeration may result in a more homogeneous and lighter surface, thereby increasing the L* value [32]. In addition, degradation or oxidation of browning-related compounds, such as Maillard reaction products, could contribute to a lighter and warmer tone [39]. Finally, starch retrogradation during cold storage could alter the surface structure of the bread, affecting light reflection and leading to increased L* and b* values [40].

### 3.2. Rheological Properties of Flours

The pasting behaviour of rice flour and flour blends prepared by substituting rice flour with almond bagasse powder, hot air-dried at 60 °C or lyophilised, is presented in Figure 1. The parameters evaluated and collected in the nested table include peak viscosity (PV), trough viscosity (TV), breakdown viscosity (BV), final viscosity (FV), setback viscosity (SV), peak time, and pasting temperature.

In general terms, the results indicate that the incorporation of almond bagasse powder, whether lyophilised or hot air-dried, affects the viscosity profile regardless of the percentage of substitution, suggesting an increase in the structural stiffness of the blends.

Nevertheless, while substitution level was the dominant factor for most viscosity parameters, the drying method also had a minor but significant effect on the gelatinisation temperature.

Although lower viscosity has been reported to facilitate product handling during processing, it may also alter the textural and sensory properties of the final product. A detailed examination of the results revealed that pasting properties such as peak viscosity (PV), trough viscosity (TV), final viscosity (FV), and setback viscosity (SV) decreased markedly in the mixtures with 25% and 30% substitution with almond bagasse powder, regardless of the drying method employed, compared to the control. This reduction could be attributed to the higher fibre content present in the almond bagasse powder [6,13], which reduces the WAC as shown in the values presented in Table 1, thereby reducing the availability of free water necessary for the gelatinisation of rice starch. Conversely, mixtures with 5% and 10% substitution exhibited higher values of trough viscosity (TV), final viscosity (FV), and setback viscosity (SV) when compared to the control. It could be explained by a higher availability of water for the hydration of starch derived from rice, which promotes amylose release [41]. At these lower concentrations, the moderate presence of fibre from the almond bagasse powder does not significantly interfere with starch hydration, thus preserving its gelatinisation capacity. However, as the substitution level increases to 25% and 30%, the amount of starch derived from rice flour in the mixture decreases considerably. The observed differences between lyophilised (LYO) and hot air-dried (HAD60) almond bagasse powders in terms of pasting viscosities can be attributed to several structural and physical factors. Lyophilisation produces powders with significantly higher porosity and larger specific surface area compared to hot air drying [33,42]. This enhanced porosity enables LYO powder to absorb water more rapidly and extensively, creating stronger competition with rice starch for available water during the pasting process. Consequently, the reduced water availability for starch gelatinisation results in lower peak, trough, and final viscosities in LYO-containing samples.

Additionally, the preservation of native protein structure in lyophilised powder (due to the absence of thermal denaturation during freeze-drying) may result in different protein–starch interactions compared to HAD60 powder, where thermal treatment at 60 °C could modify protein functionality [31]. The more compact, less porous structure of HAD60 powder absorbs water more gradually, allowing for more controlled starch hydration and consequently higher viscosity development. These structural differences explain why, despite both powders reducing overall viscosity compared to rice flour alone, LYO consistently shows lower values than HAD60 across most substitution levels.

In these cases, the fibre present in the almond bagasse powder begins to compete for the available water, thereby reducing the amount of water accessible for starch gelatinisation. Moreover, the water–fibre interaction further limits starch hydration, negatively impacting the pasting properties. As a result, mixtures with higher substitution levels exhibit a reduction in viscosity during the test, reflecting a diminished ability to form gels and undergo starch gelatinisation. Similar findings have been reported in previous studies [43]. The almond bagasse substitution increases (5% to 30%), and the total dietary fibre content in the flour blends progressively increases, creating a systematic cascade of effects on pasting behaviour. The primary mechanism involves fibre–water competition, where increasing fibre content reduces the water available for starch gelatinisation in a dose-dependent manner. Specifically, this fibre-mediated water competition influences pasting parameters as peak viscosity (PV) decreases with higher fibre content because less water is available for initial starch swelling and amylose leaching. Trough viscosity (TV) similarly decreases as fibre continues to sequester water during the high-temperature holding phase. Final viscosity (FV) and setback viscosity (SV) show the most dramatic reductions at ≥25% substitution levels, where high fibre content severely restricts the water needed for amylose retrogradation and gel network formation during cooling. Breakdown viscosity (BV), representing the difference between peak and trough viscosity, also decreases because the limited starch gelatinisation results in fewer swollen granules available for mechanical breakdown.

This systematic fibre-induced reduction in pasting properties becomes particularly pronounced at substitution levels ≥ 25%, where the fibre content reaches a critical threshold that fundamentally alters the starch–water dynamics, shifting the system from starch-dominated to fibre-dominated rheological behaviour [41]. Understanding this relationship is crucial for predicting how almond bagasse incorporation will affect bread texture and processing characteristics.

Figure 2 presents the analysis of the viscoelastic properties of the mixtures incorporating almond bagasse powder. For this purpose, a dynamic oscillatory test was carried out to determine the storage or elastic modulus (G′), the loss or viscous modulus (G′′), and the loss tangent (tan δ = G′′/G′). The latter represents the ratio between the viscous and elastic responses, which defines the mechanical behaviour of the material [44]. It was observed that, in all cases, the values of the storage or elastic modulus (G′) were higher than those of the loss or viscous modulus (G′′), indicating a predominance of the elastic component over the viscous one. The highest values of G′ and G″ were recorded in the control sample, whereas a reduction was observed in those formulations where rice flour was partially replaced with almond bagasse powder, lyophilised or hot air-dried. This highlights the impact of the substitution on both viscoelastic moduli, as the addition of almond bagasse powder leads to a decrease in both G′ and G″. This reduction could be attributed to the proportion of protein and dietary fibre present in almond bagasse, as these components contribute to a denser and more cohesive structure, which influences their viscoelastic properties [45]. Furthermore, the values of the loss tangent remained below 0.42 across all samples and varied slightly with increasing frequency. According to studies by Fetouhi et al. [45], a tan δ value below 1 in gluten-free doughs is associated with a solid-like structural behaviour. Similar findings have been reported in products formulated with rice flour [46].

### 3.3. Textural Characteristics of Gels and Bread Under Fresh and Stored Conditions

This section examines how the incorporation of almond bagasse powder influences the firmness, elasticity, and structure of the resulting gels. As shown in Figure 3, gels prepared with rice flour and increasing levels of substitution (5–30%) with almond bagasse powder—lyophilised or hot air-dried—exhibited significant differences in their textural behaviour. In particular, the gel formulated exclusively with rice flour (control) exhibited the highest maximum force during both the first and second compressions. As the substitution level increased, the force required to compress the gels progressively decreased. This means that the incorporation of almond bagasse powder negatively affects the firmness and structural strength of the gels, which can be mainly attributed to a reduction in the starch content available for gelatinisation during the pasting process. Rice flour, rich in starch, can form a strong structural network when heated in the presence of water. In contrast, almond bagasse powder, which is low in starch and high in dietary fibre, limits the formation of this network, resulting in softer gel textures [47]. Moreover, in all treatments, the force recorded during the second compression was systematically lower than the first. This difference suggests a reduced ability of the gels to recover their structure after deformation, which is associated with decreased cohesiveness and elasticity of the system. In gelled products, a reduction in force during the second compression indicates a less elastic matrix, particularly when the proportion of components that do not contribute to network formation—such as bagasse fibre—increases [43].

On the other hand, gels prepared with blends containing hot air-dried almond bagasse powder exhibited slightly higher resistance to deformation compared to those formulated with lyophilised powder. This difference could be attributed to the fact that hot air drying produces denser and less porous particles, which may favour the formation of a more compact gel matrix. In addition, thermal treatment may induce modifications in proteins or polysaccharides that promote more rigid interactions within the gel structure [48]. The lower Water Absorption Capacity observed in hot air-dried powders, compared to lyophilised ones, may also have contributed to the formation of gels with reduced water content and increased firmness [31,42].

The textural behaviour of bread can be significantly affected by the inclusion of fibre-rich ingredients. In this context, the effect of almond bagasse powder substitution on bread firmness and cohesiveness was evaluated, considering both in the fresh state (Figure 4a) and after 7 days of refrigerated storage (Figure 4b). In the case of fresh bread, it was observed that the bread made with rice flour (control) required greater force during the first compression compared to those containing almond bagasse powder. This can be explained by the fact that the high starch content in rice flour allows the formation of a dense and firm structural network during baking. As this network is not disrupted by other components, it results in a more compact crumb that requires a greater effort to compress [47]. Conversely, the incorporation of almond bagasse powder interferes with the formation of the starch network due to its fibre content, leading to a less dense and firm structure, and consequently requiring less force to compress [40]. This effect was even more pronounced in breads made with lyophilised almond bagasse powder compared to those containing hot air-dried powder. Lyophilisation produces a porous structure capable of absorbing water, which results in a moister crumb and a softer texture [31]. When the second compression was evaluated, a reduced structural recovery was observed in all breads, indicating lower elasticity. As with the first compression, this behaviour is associated with the interference of almond bagasse powder in the formation of the starch network during baking, thereby limiting the bread’s ability to regain its shape after deformation [49]. This reduction in elasticity was more evident in breads made with lyophilised bagasse powder compared to those containing hot air-dried powder [31]. These findings support the idea that the softer texture observed in breads with lyophilised bagasse is related to a less elastic and more fragile internal structure.

The texture was also affected by refrigerated storage for 7 days (Figure 4b). In this case, all treatments showed a clear hardening compared to fresh bread, particularly the control sample. Starch retrogradation, a typical phenomenon during cold storage, in which amylose and amylopectin chains tend to reorganise into more ordered and crystalline structures, would be the main mechanism responsible for these changes. This reorganisation involves water release from the bread matrix (syneresis), resulting in a drier, harder, and more brittle crumb [50]. This structural change was reflected in the ease with which stored breads fractured during the first compression, especially the control and the LYO-5% treatment, which recorded the highest force values. This fracturing behaviour during the first compression introduces an important methodological limitation for TPA analysis of highly brittle samples. When samples fracture during initial compression, parameters derived from the second compression—particularly cohesiveness (calculated as the ratio of work conducted during the second to first compression)—become less reliable and not directly comparable to those obtained from intact, less brittle samples. Therefore, for the most brittle stored samples (particularly the control and LYO-5%), the second compression data should be interpreted with caution, and direct comparisons of cohesiveness values between highly brittle and more elastic samples may not be entirely valid. The first compression parameters (maximum force, firmness) remain reliable indicators of sample hardness regardless of sample integrity. The increased fragility observed in both cases can be attributed to their high starch content and the enhanced interaction capacity of lyophilised powder with the bread matrix [39]. Although this type of powder has a higher water retention capacity, the low substitution level in the LYO-5% treatment was insufficient to generate a significant structural difference compared to the control. The limited presence of fibre or components with softening effects also contributed to a more brittle texture. As a result, the second compression was performed on an already fractured bread sample, which affected the force values recorded. This behaviour aligns with previous findings, which identify starch retrogradation as the main cause of hardening in gluten-free breads during cold storage [39,51].

### 3.4. Antioxidant Properties and Reducing Sugar Content of Flour and Bread Under Fresh and Stored Conditions

Figure 5 shows the results for antioxidant activity (DPPH in Figure 5a, FRAP in Figure 5b), total phenol content (Figure 5c), and reducing sugars (Figure 5d) in rice flour (control) and rice flour blends with lyophilised almond bagasse (LYO) or hot air-dried almond bagasse at 60 °C (HAD60), and substitution levels of 5%, 10%, 15%, 20%, 25%, and 30%. Additionally, the control, made exclusively from rice flour, was analysed.

A significant increase in antioxidant capacity determined by DPPH assay was observed as the substitution level with almond bagasse increased (Figure 5a). However, the blends including lyophilised powder (LYO) showed lower antioxidant capacity compared to those with hot air-dried powder (HAD60). The HAD60-30% sample exhibited the highest antioxidant activity, significantly surpassing both the control and other formulations. It can be attributed to the formation of secondary products with high antioxidant activity, primarily generated during the hot air-drying process through the Maillard reaction. As the substitution level increases, the compounds involved in this reaction also increase, contributing to a greater antiradical capacity [37,52,53]. Furthermore, thermal processing could enhance the release of antioxidant compounds previously bound to the food matrix, thereby increasing their availability [54].

Figure 5b shows the results of the FRAP assay, which measures the ferric reducing capacity as another method to assess antiradical capacity. Again, a trend of increasing antiradical capacity was observed as the substitution level increased, especially in the blends made with hot air-dried almond bagasse (HAD60). While no significant differences were found between the HAD60 and LYO samples at substitution levels of 5% to 25%, the HAD60-30% sample showed a substantial increase in its reducing capacity. This result suggests that at higher substitution levels, the thermal treatment applied to the bagasse could favour the formation or release of antioxidant compounds, probably derived from browning reactions such as the Maillard reaction [55]. On the other hand, the control made exclusively with rice flour exhibited the lowest antioxidant activity, which is consistent with its low phenolic compound content and the absence of thermal treatments that promote the formation of additional antioxidants [56].

The results for total phenolic content are presented in Figure 5c. Rice flour (control) exhibited the lowest value, whereas the blends containing almond bagasse powder showed significant increases. This enhancement can be attributed to the incorporation of almond bagasse, a material naturally rich in phenolic compounds. In the formulations prepared with hot air-dried powder (HAD60), the phenolic content progressively increased with the level of substitution, reaching its highest value in the HAD60-30% sample. The blends incorporating lyophilised powder (LYO) exhibited even higher phenolic contents, particularly at the 25% and 30% substitution levels. This higher phenolic content could be explained considering that hot air drying can induce partial degradation of phenolic compounds due to thermal exposure [57], whereas lyophilisation more effectively preserves their chemical stability by avoiding high temperatures [58].

Finally, regarding the analysis of the flours, the results corresponding to the reducing sugars content are presented in Figure 5d. These compounds, characterised by the presence of a free aldehyde or ketone group, not only influence the sensory properties of foods but also contribute to their antioxidant capacity. Rice flour (control) exhibited the lowest reducing sugar content. The incorporation of almond bagasse powder, lyophilised or hot air-dried, resulted in a significant increase in all blends evaluated. This increase was more pronounced in the formulations containing lyophilised powder, particularly in LYO-25% and LYO-30%, which recorded the highest values. It can be attributed to the natural content of soluble sugars in almond bagasse [59] and to the ability of lyophilisation to better preserve these compounds, minimising losses associated with non-enzymatic browning reactions, such as the Maillard reaction, which can occur during hot air drying [33].

After evaluating the antioxidant properties and reducing sugar content of the flours, their levels in breads prepared from blends of rice flour and almond bagasse powder were subsequently analysed both in fresh and stored conditions (7 days in refrigeration). These analyses provided key information on how the properties of the flours influenced the antioxidant characteristics of the breads over time.

Figure 6a, corresponding to the DPPH analysis, and Figure 6b, corresponding to the FRAP analysis, show that the antioxidant capacity was higher in the crust than in the crumb of all the breads, including those made with rice flour only (control). This behaviour is attributed to the formation of antioxidant compounds due to Maillard reactions during baking, intensified in areas exposed to high temperatures [55,60]. Similar results were reported for rye bread, where greater antioxidant activity was also observed in the crust compared to the crumb [61,62]. Breads made with almond bagasse blends showed increased antioxidant capacity (DPPH and FRAP) with higher substitution levels, especially when using hot air-dried powder (HAD60), likely due to the combined effects of thermal drying and baking [63,64]. Lyophilised powder (LYO) also improved antioxidant activity at higher levels, though to a lesser extent. This suggests that while freeze-drying better preserves antioxidants, hot air drying and baking promote the formation of additional compounds with antioxidant potential [65]. The antioxidant enrichment becomes more evident at 25–30% substitution in both cases.

The total phenolic content (Figure 6c) showed significant differences between the various formulations. In general, a progressive increase in phenolic concentration was observed as the substitution level with almond bagasse powder, hot air-dried (HAD60) or lyophilised (LYO), increased. In all cases, the values were higher in the crust compared to the crumb, in line with previous reports indicating that baking promotes surface transformations in bread that increase the concentration of phenolic compounds [55]. In breads, phenolic content increased with higher levels of almond bagasse substitution, particularly at 25% and 30%. Notably, hot air-dried powder (HAD60) led to greater phenolic release, especially at high substitution levels, likely due to thermal pre-treatment enhancing compound availability during baking [63]. Lyophilised powder (LYO), while preserving phenolics, showed a slower release. These results underscore the impact of drying method on phenolic compound behaviour under baking conditions. The control bread, made only with rice flour, exhibited the lowest total phenolic values, in agreement with the results obtained in the DPPH and FRAP assays, as well as with those reported by Rocchetti et al. [66].

To conclude the evaluation of the fresh breads in terms of reducing sugar content (Figure 6d), significant differences were found in both crumb and crust. In general, reducing sugar levels were higher in the crust, probably due to the effect of high temperatures during baking, which promote the concentration and transformation of simple sugars at the bread surface [67]. The bread made with rice flour (control) exhibited the lowest levels of reducing sugars, which can be attributed to its naturally low content of soluble sugars [68]. On the other hand, although the flour blends incorporating the lyophilised powder had more reducing sugars (Figure 5d), breads incorporating hot air-dried powder (HAD60) showed higher levels of reducing sugars than those prepared with lyophilised powder (LYO), particularly at substitution levels from 15% to 30%. This difference could be attributed to the increased structural damage caused by lyophilisation (generally higher porosity) in the powder preparation. It may cause the reducing sugars to be consumed more rapidly during the Maillard reactions that occur during baking, while the HAD60 powder, having already undergone some thermal treatment, might release sugars more slowly or differently [69].

After analysing the characteristics of fresh breads, this study continued with the evaluation of breads stored in refrigeration for 7 days, with the aim of assessing the stability of antioxidant properties, total phenolic content, and reducing sugars over time.

During the storage time, antioxidant activity, assessed using the DPPH (Figure 7a) and FRAP (Figure 7b) methods, along with total phenolic content (Figure 7c), maintained the same trend observed in fresh breads: higher values in the crust compared to the crumb, as well as progressive increases as the substitution level with almond bagasse (both hot air-dried and lyophilised) increased. The control bread continued to show the lowest values in all assays. However, refrigerated storage resulted in a general decrease in antioxidant activity and total phenolic content, which can be attributed to oxidation and degradation of phenolic compounds during storage [70]. When comparing fresh and stored breads, it was observed that losses of these compounds were smaller as the level of substitution with almond bagasse increased, regardless of the drying method used. The magnitude of this reduction depended on the type of drying treatment applied to the almond bagasse used in bread preparation. In the case of antioxidant activity measured by DPPH, the losses were 52.31% in HAD60-5% and 19.57% in HAD60-30%, while in breads made with lyophilised powder, losses were 81.98% in LYO-5% and 8.92% in LYO-30%. Similarly, antioxidant capacity assessed by FRAP showed losses of 13.11% in HAD60-5% and 4.31% in HAD60-30%, compared to 9.49% in LYO-5% and 5.65% in LYO-30%. In terms of total phenolic content, losses reached 29.60% in HAD60-5% and 17.50% in HAD60-30%, while in breads made with lyophilised bagasse powder, the losses were 35.68% in LYO-5% and 22.80% in LYO-30%. Despite these decreases, breads with higher levels of substitution with hot air-dried bagasse (HAD60-25% and HAD60-30%) retained the highest antioxidant capacities and phenolic contents. In contrast, breads made with lyophilised bagasse, even at high substitution levels (LYO-25% and LYO-30%), exhibited more significant losses, which suggests a greater susceptibility of their phenolic compounds to cold storage conditions [71].

Regarding the reducing sugar content (Figure 7d) in the stored breads, formulations incorporating almond bagasse powder, both hot air-dried and lyophilised, showed the same trend as in the fresh breads. Higher concentrations were recorded compared to the control bread, with a progressive increase as the substitution level rose. This trend was evident in both the crust and the crumb. After seven days of refrigeration, a general decrease in reducing sugar content was detected, although of a small magnitude. This loss could be related to the participation of these sugars in Maillard-type reactions during storage, as well as to degradation or transformation processes associated with bread ageing [72].

Finally, Table 3 summarises the percentage increases observed in antioxidant activity (DPPH and FRAP), total phenolic content, and reducing sugars in fresh and 7-day stored breads, made with blends of rice flour and almond bagasse powder (HAD60 or LYO) at different substitution levels (5–30%) compared to the control bread. These results highlight the following key points:
The increases observed in the fresh bread with almond bagasse substitution relative to the control are mainly attributed to the contribution of bioactive compounds and sugars present in the almond bagasse, as well as to Maillard reactions generated during the thermal pre-treatment of the bagasse powder and the bread baking process. These factors promote the release or transformation of phenolic compounds, enhancing the functional properties of the bread.Breads stored for seven days and formulated with hot air-dried bagasse (HAD60) showed more pronounced increases, possibly because antioxidant compounds continue to form during storage as a result of ongoing Maillard reactions, further enhancing the bread’s functional properties. Therefore, the drying method clearly influences the stability and retention of these bioactive compounds during storage.


### 3.5. Potential Strategies for Texture Optimisation

While this study demonstrates that almond bagasse incorporation significantly enhances the nutritional profile of gluten-free bread, the observed reduction in firmness and elasticity, particularly at substitution levels ≥ 20%, presents challenges for commercial application. Based on our findings and established food science principles, several strategies could potentially mitigate these texture limitations, like hydrocolloid addition, particle size optimisation, flour blend optimisation, enzymatic optimisation, hydration adjustment, or process modification.

The incorporation of hydrocolloids such as xanthan gum, guar gum, or hydroxypropylmethylcellulose (HPMC) could compensate for the weakened starch network [47]. Given that our results show reduced viscoelastic moduli (Figure 2) and compromised gel strength (Figure 3) with almond bagasse addition, hydrocolloids could restore structure-building capacity and improve gas retention during fermentation.

Our study used powder milled to pass through 500 μm sieves, but finer milling could improve functionality. Reducing almond bagasse particle size could enhance water distribution, reduce fibre interference with starch gelatinisation, and improve overall dough cohesiveness [34]. This approach could be particularly effective given our pasting results showing water competition effects.

Combining almond bagasse with protein-rich flours (e.g., chickpea, quinoa) or incorporating vital wheat gluten analogues could strengthen the bread matrix [27]. Our rheological data suggest that protein addition could help restore the elastic properties lost due to starch network disruption.

While this study maintained constant hydration to assess ingredient impact, future optimisation could adjust water content based on the specific WAC of each blend (Table 1). This could partially compensate for the water competition effects we observed in pasting behaviour.

Targeted enzyme treatments (e.g., transglutaminase for protein cross-linking or specific amylases for starch modification) could enhance structural integrity while preserving the nutritional benefits of almond bagasse incorporation.

Extended fermentation times or modified mixing protocols could allow better hydration of fibre components and improved dough development, potentially addressing some texture limitations observed in our texture analysis.

These strategies warrant systematic investigation in future studies to optimise the balance between nutritional enhancement and textural quality, potentially enabling higher substitution levels while maintaining consumer acceptability.

## 4. Conclusions

The partial enrichment of rice flour with almond bagasse powder (in the range of 5–30%) in gluten-free breads significantly changed their physicochemical, functional, and nutritional properties of gluten-free bread, with more pronounced effects as the substitution level increased. This incorporation enhanced moisture retention, reduced firmness, and improved crumb texture. The drying method of the almond bagasse had a notable impact on product quality: lyophilisation resulted in breads with greater specific volume and a more aerated structure, whereas hot-air drying promoted Maillard reactions, thereby increasing antioxidant capacity, particularly in the crust. After seven days of storage, all breads hardened due to starch retrogradation, with some samples becoming sufficiently brittle to affect the reliability of texture analysis parameters dependent on sample integrity. However, those with higher substitution levels showed reduced antioxidant losses, with greater oxidative stability observed in formulations containing hot air-dried almond bagasse compared to the lyophilised counterpart. Overall, almond bagasse stands out as a functional and sustainable ingredient that adds value to an agro-industrial by-product while enhancing the quality and shelf-life of gluten-free products. The choice of drying method proves to be a key factor in guiding the technological and nutritional profile of the final product. Future research should focus on texture optimisation strategies, including hydrocolloid addition and particle size optimisation, to enable higher substitution levels while maintaining desirable bread characteristics.

## Figures and Tables

**Figure 1 foods-14-02382-f001:**
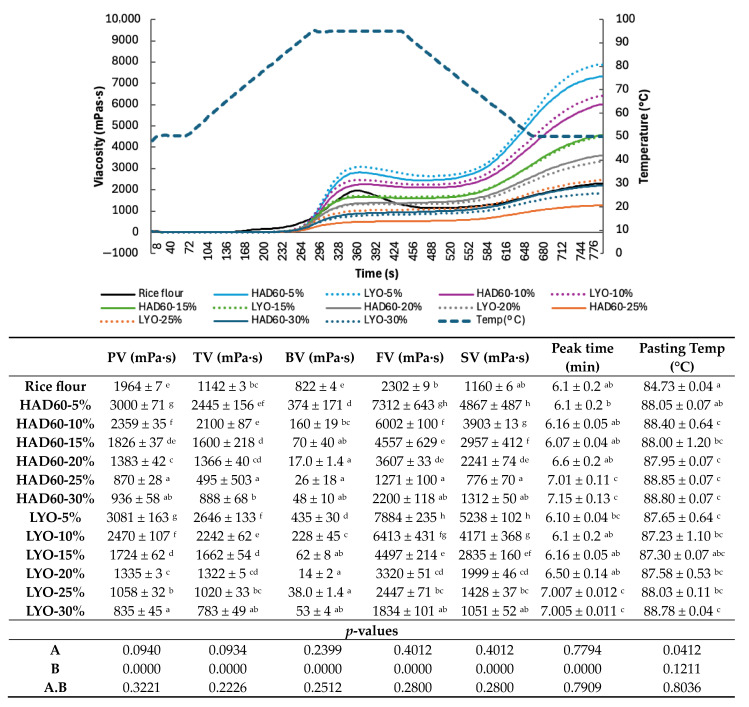
Pasting curves, pasting temperature, and the pasting behaviour of rice flour (control) and flour blends in which rice flour was partially replaced with almond bagasse powder, hot air-dried at 60 °C (HAD60) or lyophilised (LYO), at substitution levels of 5–30%. PV: peak viscosity; TV: trough viscosity; BV: breakdown viscosity; FV: final viscosity; SV: setback viscosity. (A) Treatment; (B) replacement percentage. The results represent the mean of three repetitions. Different lowercase letters indicate significant differences (*p* ≤ 0.05) among the different flours.

**Figure 2 foods-14-02382-f002:**
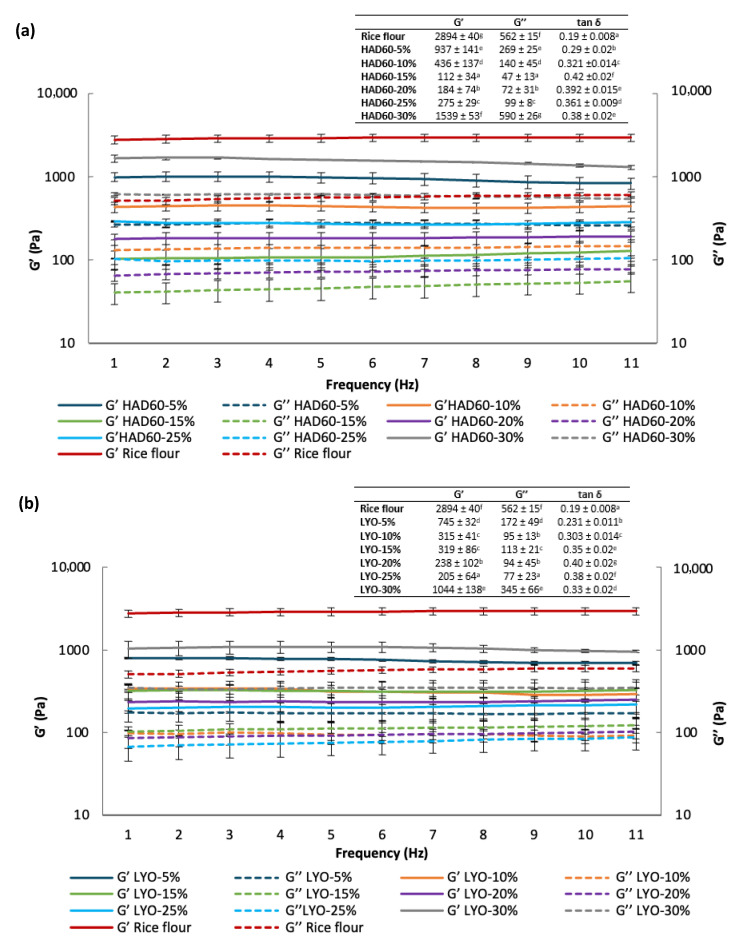
Elastic modulus (G′) and viscous modulus (G″) of rice flour (control) and of flour blends in which rice flour was partially replaced with almond bagasse powder, hot air-dried at 60 °C (HAD60) (**a**) or lyophilised (LYO) (**b**), at substitution levels of 5–30%. The results represent the mean of three repetitions. Different lowercase letters indicate significant differences (*p* ≤ 0.05) among the different flours.

**Figure 3 foods-14-02382-f003:**
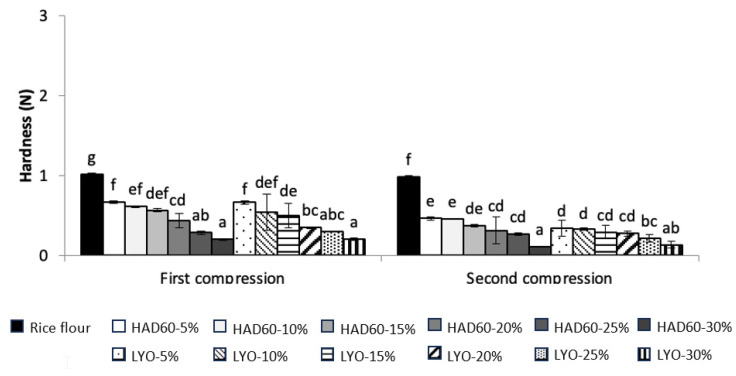
Gel hardness (N) of samples prepared from rice flour (control) and from flour blends in which rice flour was partially replaced with almond bagasse powder, hot air-dried at 60 °C (HAD60) or lyophilised (LYO), at substitution levels of 5–30%. All samples were hydrated, heated, and cooled to allow gel formation prior to analysis. The results represent the mean of three repetitions. Different lowercase letters indicate significant differences (*p* ≤ 0.05) among the different flours.

**Figure 4 foods-14-02382-f004:**
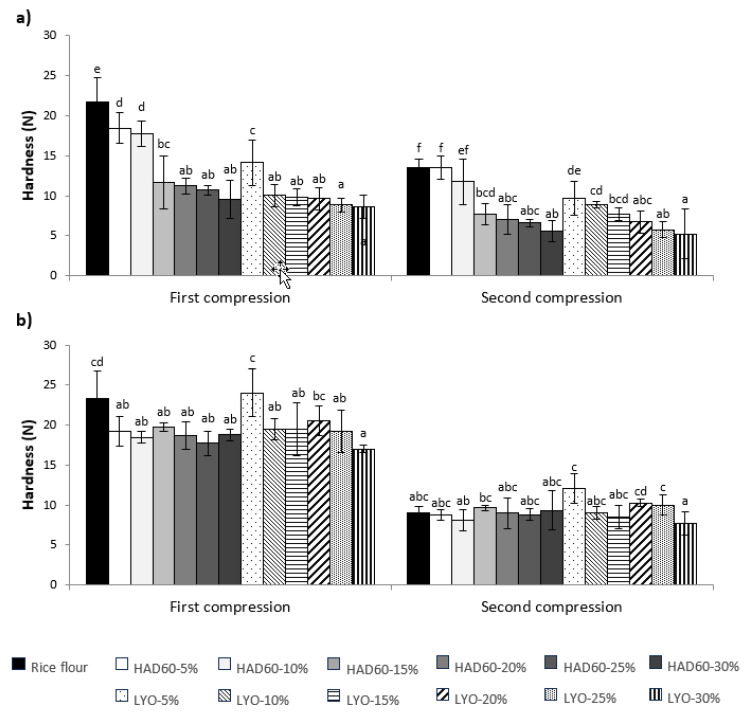
Hardness (N) of fresh (**a**) and 7-day stored (**b**) bread made from rice flour (control) and flour blends in which rice flour was partially replaced with almond bagasse powder, hot air-dried at 60 °C (HAD60) or lyophilised (LYO), at substitution levels of 5–30%. The results represent the mean of three repetitions. Different lowercase letters in each compression indicate significant differences (*p* ≤ 0.05) among the different bread formulations.

**Figure 5 foods-14-02382-f005:**
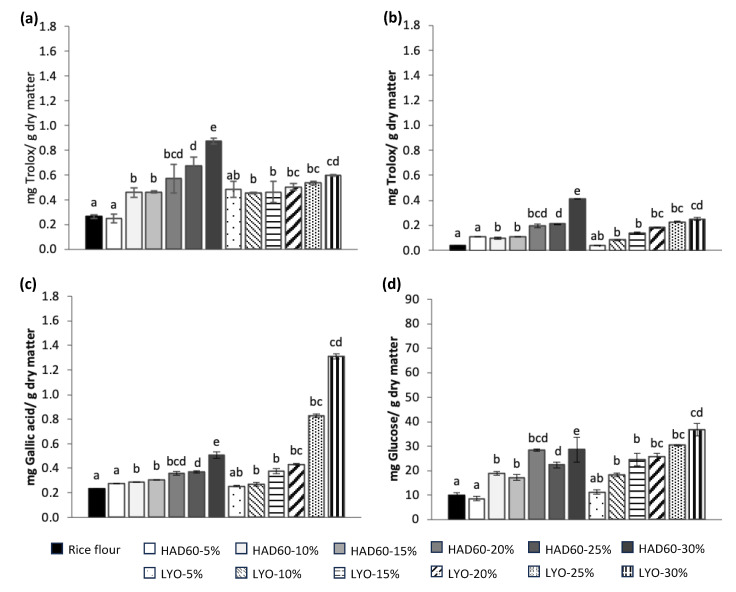
Antioxidant activity determined by (**a**) DPPH and (**b**) FRAP methods; (**c**) total phenol content; and (**d**) reducing sugars in samples prepared from rice flour (control) and from flour blends in which rice flour was partially replaced with almond bagasse powder, hot air-dried at 60 °C (HAD60) or lyophilised (LYO), at substitution levels of 5–30%. The results represent the mean of three repetitions. Different lowercase letters indicate significant differences (*p* ≤ 0.05) among the different flours.

**Figure 6 foods-14-02382-f006:**
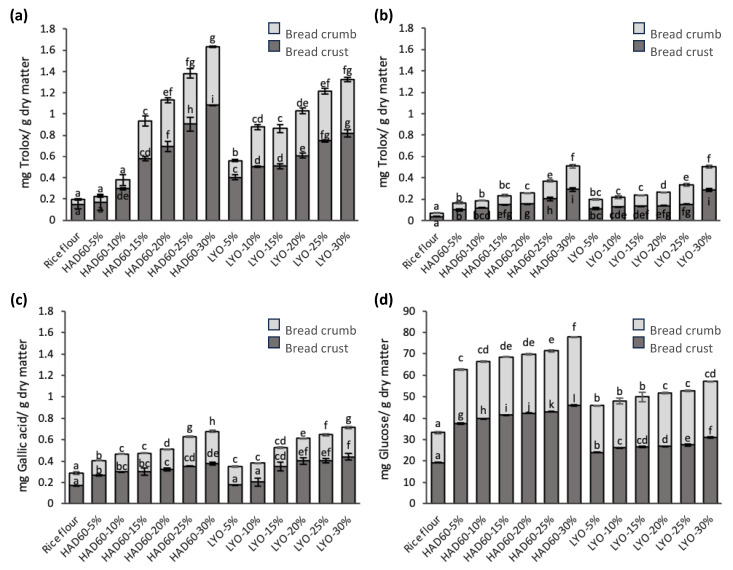
Antioxidant activity determined by (**a**) DPPH and (**b**) FRAP methods; (**c**) total phenol content; and (**d**) reducing sugars in the crumb and crust of fresh bread made from rice flour (control) and flour blends in which rice flour was partially replaced with almond bagasse powder, hot air-dried at 60 °C (HAD60) or lyophilised (LYO), at substitution levels of 5–30%. The results represent the mean of three repetitions. Different lowercase letters within each bread fraction (crumb or crust) indicate significant differences (*p* ≤ 0.05) among the different bread formulations.

**Figure 7 foods-14-02382-f007:**
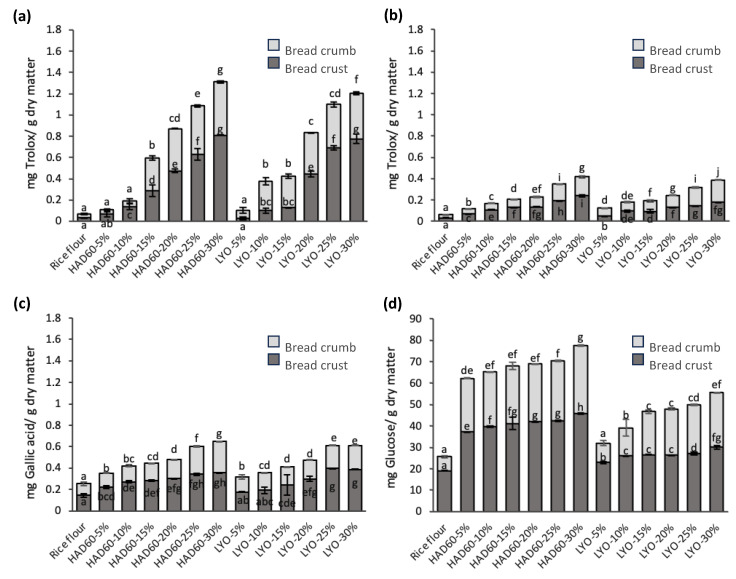
Antioxidant activity determined by (**a**) DPPH and (**b**) FRAP methods; (**c**) total phenol content; and (**d**) reducing sugars in the crumb and crust of 7-day stored bread made from rice flour (control) and flour blends in which rice flour was partially replaced with almond bagasse powder, hot air-dried at 60 °C (HAD60) or lyophilised (LYO), at substitution levels of 5–30%. The results represent the mean of three repetitions. Different lowercase letters within each bread fraction (crumb or crust) indicate significant differences (*p* ≤ 0.05) among the different bread formulations.

**Table 1 foods-14-02382-t001:** Physicochemical and techno-functional properties of rice flour, almond bagasse powders dehydrated by hot air at 60 °C (HAD60) and by lyophilisation (LYO), and flour blends in which rice flour was partially replaced with almond bagasse powder at substitution levels of 5–30%.

Sample	Xw (g Water/g)	WAC (g Water/g)	OAC (g Oil/g)	Colour				
				L*	a*	b*	C*	∆E
LYO ^1^	0.02 ± 0.08	8.4 ± 1.8	4.20 ± 0.06	66.56 ± 0.01	6.039 ± 0.002	15.026 ± 0.014	16.190 ± 0.012	6.7 ± 0.2
HAD60 ^1^	0.014 ± 0.002	2.9 ± 0.5	2.3 ± 0.5	62.358 ± 0.010	4.999 ± 0.009	14.279 ± 0.006	15.128 ± 0.08	9.8 ± 0.5
Rice flour	0.1228 ± 0.0010 ^g^	2.25 ± 0.03 ^f^	1.57 ± 0.03 ^ab^	56.6 ± 0.8 ^a^	2.655 ± 0.006 ^i^	10.5 ± 0.3 ^d^	11.20 ± 0.07 ^b^	-
HAD60-5%	0.0968 ± 0.0004 ^a^	2.13 ± 0.04 ^def^	1.60 ± 0.03 ^abc^	115 ± 2 ^g^	1.809 ± 0.011 ^b^	8.95 ± 0.14 ^a^	9.1 ± 0.2 ^a^	3 ± 2 ^b^
HAD60-10%	0.0921 ± 0.0008 ^b^	2.06 ± 0.07 ^bcd^	1.56 ± 0.11 ^ab^	112.4 ± 1.3 ^efg^	1.820 ± 0.012 ^b^	9.09 ± 0.09 ^a^	9.13 ± 0.13 ^a^	3.1 ± 0.2 ^b^
HAD60-15%	0.087 ± 0.002 ^c^	2.08 ± 0.07 ^bcde^	1.53 ± 0.10 ^ab^	113.8 ± 1.0 ^fg^	1.88 ± 0.02 ^c^	9.47 ± 0.08 ^b^	9.65 ± 0.08	3.5 ± 0.4
HAD60-20%	0.0848 ± 0.0004 ^d^	2.10 ± 0.11 ^cde^	1.55 ± 0.10 ^ab^	111.2 ± 1.5 ^def^	2.015 ± 0.040 ^d^	9.7 ± 0.2 ^bc^	10.2 ± 0.2 ^ab^	5 ± 2 ^d^
HAD60-25%	0.0812 ± 0.0002 ^e^	1.97 ± 0.08 ^ab^	1.52 ± 0.02 ^a^	109.3 ± 1.1 ^cd^	2.247 ± 0.023 ^f^	10.6 ± 0.3 ^d^	10.27 ± 0.27 ^ab^	6 ± 2 ^d^
HAD60-30%	0.0771 ± 0.0004 ^f^	1.99 ± 0.09 ^abc^	1.59 ± 0.03 ^abc^	106.2 ± 2.3 ^b^	2.55 ± 0.04 ^h^	11.2 ± 0.2 ^e^	8.9 ± 5.1 ^a^	7 ± 2 ^de^
LYO-5%	0.09554 ± 0.00014 ^a^	2.182 ± 0.007 ^ef^	1.73 ± 0.08 ^d^	111.4 ± 2.6 ^def^	1.70 ± 0.03 ^a^	8.9 ± 0.2 ^a^	9.0 ± 0.2 ^a^	2.3 ± 0.2 ^a^
LYO-10%	0.09196 ± 0.00006 ^a^	2.135 ± 0.003 ^def^	1.69 ± 0.08 ^cd^	112.5 ± 0.3 ^efg^	1.80 ± 0.02 ^b^	8.9 ± 0.2 ^a^	9.06 ± 0.01 ^a^	3.0 ± 0.3 ^b^
LYO-15%	0.0876 ± 0.0004 ^c^	2.04 ± 0.02 ^abcd^	1.63 ± 0.06 ^bcd^	110.0 ± 2.2 ^de^	2.21 ± 0.02 ^e^	10.4 ± 0.2 ^d^	10.6 ± 0.2 ^ab^	3.1 ± 0.2 ^b^
LYO-20%	0.0837 ± 0.0007 ^d^	2.0 ± 0.2 ^abcd^	1.58 ± 0.02 ^ab^	112.6 ± 1.7 ^fg^	2.370 ± 0.01 ^g^	10.5 ± 0.2 ^d^	10.8 ± 0.2 ^ab^	3.3 ± 0.6 ^b^
LYO-25%	0.077 ± 0.003 ^e^	2.00 ± 0.02 ^abc^	1.53 ± 0.03 ^ab^	109.6 ± 3.5 ^d^	2.37 ± 0.05 ^g^	10.0 ± 0.3 ^c^	10.3 ± 0.3 ^ab^	3.4 ± 0.3 ^bc^
LYO-30%	0.076 ± 0.002 ^f^	1.94 ± 0.03 ^a^	1.54 ± 0.07 ^ab^	107.0 ± 1 ^bc^	2.378 ± 0.003 ^g^	10.5 ± 0.3 ^d^	10.40 ± 0.07 ^ab^	3.6 ± 0.4 ^c^

The results represent the mean of three repetitions. Different lowercase letters indicate significant differences (*p* < 0.05) among the different flours. Xw: moisture; WAC: Water Absorption Capacity; OAC: Oil Absorption Capacity; L*a*b*C*: colour parameters; and ∆E: colour differences ^1^ [6].

**Table 2 foods-14-02382-t002:** Physicochemical properties of fresh and 7-day stored bread made from rice flour (control) and flour blends in which rice flour was partially replaced with almond bagasse powder, hot air-dried at 60 °C (HAD60) or lyophilised (LYO), at substitution levels of 5–30%.

Sample	aw	Specific Volume (cm^3^/g)	Relative Weight	Colour				
				L*	a*	b*	C*	∆E
Fresh bread								
Rice flour	0.985 ± 0.002 ^a^	2.34 ± 0.02 ^e^	-	57.2 ± 1.3 ^de^	−4.06 ± 0.14 ^a^	10.9 ± 0.2 ^bcdfg^	11.6 ± 0.2 ^e^	-
HAD60-5%	0.992 ± 0.006 ^b^	2.38 ± 0.03 ^ef^	1.02 ± 0.08 ^a^	57.1 ± 2.3 ^de^	−3.7 ± 0.4 ^ab^	9.9 ± 0.6 ^a^	10.5 ± 0.6 ^a^	2 ± 3 ^ab^
HAD60-10%	0.989 ± 0.005 ^ab^	2.586 ± 0.011 ^h^	1.00 ± 0.05 ^a^	58.1 ± 3.4 ^cd^	−3.1 ± 0.5 ^bc^	11.0 ± 0.6 ^dfg^	11.5 ± 0.5 ^e^	2 ± 4 ^ab^
HAD60-15%	0.9890 ± 0.0010 ^ab^	2.05 ± 0.03 ^cd^	1.02 ± 0.06 ^a^	54.4 ± 3.1 ^abc^	−2.4 ± 0.5 ^d^	10.6 ± 0.4 ^bcd^	10.9 ± 0.4 ^bc^	2 ± 3 ^ab^
HAD60-20%	0.9883 ± 0.0006 ^ab^	2.02 ± 0.03 ^bcd^	1.03 ± 0.07 ^a^	55 ± 3 ^bcd^	−2.0 ± 0.6 ^de^	10.6 ± 0.4 ^bc^	10.7 ± 0.2 ^abc^	2 ± 2 ^ab^
HAD60-25%	0.984 ± 0.003 ^a^	1.944 ± 0.005 ^ab^	1.05 ± 0.09 ^a^	53 ± 3 ^ab^	−1.8 ± 0.4 ^ef^	10.5 ± 0.4 ^cde^	10.6 ± 0.7 ^ab^	2 ± 2 ^ab^
HAD60-30%	0.987 ± 0.002 ^ab^	1.970 ± 0.006 ^abc^	1.05 ± 0.07 ^a^	52.9 ± 3.1 ^a^	−1.5 ± 0.7 ^fg^	10.9 ± 0.7 ^bc^	11.0 ± 0.3 ^c^	4 ± 2 ^b^
LYO-5%	0.986 ± 0.004 ^ab^	1.93 ± 0.12 ^ab^	1.03 ± 0.09 ^a^	55.6 ± 2.2 ^cd^	−3.1 ± 0.3 ^c^	10.4 ± 0.3 ^b^	10.9 ± 0.4 ^abc^	2 ± 2 ^a^
LYO-10%	0.990 ± 0.005 ^ab^	1.917 ± 0.004 ^ab^	1.06 ± 0.06 ^a^	55.6 ± 2.2 ^cd^	−2.0 ± 1.8 ^def^	10.7 ± 0.6 ^bcd^	11.1 ± 0.3 ^cd^	2 ± 2 ^ab^
LYO-15%	0.989 ± 0.003 ^ab^	1.875 ± 0.012 ^a^	1.06 ± 0.07 ^a^	56.6 ± 2.2 ^de^	−2.2 ± 0.4 ^de^	10.4 ± 0.3 ^b^	10.9 ± 0.4 ^abc^	2 ± 3 ^ab^
LYO-20%	0.989 ± 0.002 ^ab^	2.10 ± 0.07 ^d^	1.06 ± 0.07 ^a^	56 ± 2 ^cde^	−3.1 ± 0.3 ^c^	11.3 ± 0.4 ^gh^	11.5 ± 0.5	2 ± 2 ^ab^
LYO-25%	0.9897 ± 0.0012 ^ab^	2.453 ± 0.015 ^fg^	1.06± 0.08 ^a^	56 ± 2 ^cde^	−1.7 ± 0.5 ^efg^	11.3 ± 0.5 ^fgh^	11.4 ± 0.6 ^de^	2 ± 3 ^ab^
LYO-30%	0.989 ± 0.002 ^ab^	2.51 ± 0.10 ^gh^	1.04 ± 0.08 ^a^	55.9 ± 1.6 ^cd^	−1.2 ± 0.6 ^g^	11.4 ± 0.6 ^h^	11.6 ± 0.2 ^e^	4 ± 2 ^b^
7-day stored bread								
Rice flour	0.989 ± 0.003 ^ab^	2.40 ± 0.07 ^c^	-	56 ± 4 ^ab^	−9.3 ± 1.2 ^a^	16.5 ± 0.7 ^fg^	18.9 ± 1.2 ^e^	8.5 ± 0.8 ^c^
HAD60-5%	0.989 ± 0.006 ^abc^	2.5776 ± 0.0012 ^de^	2.46 ± 0.02 ^abc^	60 ± 5 ^c^	−6.8 ± 0.7 ^b^	15.1 ± 0.6 ^de^	16.5 ± 0.7 ^cd^	3.5 ± 9 ^ab^
HAD60-10%	0.994 ± 0.004 ^c^	2.7230 ± 0.0013 ^f^	2.58 ± 0.02 ^c^	63.4 ± 1.3 ^d^	−5.8 ± 0.4 ^b^	15.4 ± 0.7 ^de^	16.4 ± 0.6 ^cd^	3.8 ± 6.6 ^ab^
HAD60-15%	0.988 ± 0.002 ^ab^	2.30 ± 0.08 ^b^	2.25 ± 0.09 ^ab^	59.7 ± 2.4 ^bc^	−3.41 ± 0.04 ^cd^	9.8 ± 0.4 ^a^	10.5 ± 0.3 ^ab^	4 ± 4 ^ab^
HAD60-20%	0.9883 ± 0.0006 ^ab^	2.26 ± 0.04 ^b^	2.26 ± 0.05 ^ab^	59 ± 2 ^c^	−1.7 ± 0.9 ^b^	11.2 ± 0.4 ^b^	11.3 ± 0.4 ^b^	4 ± 4 ^ab^
HAD60-25%	0.985 ± 0.002 ^a^	2.26 ± 0.02 ^b^	2.26 ± 0.02 ^a^	58 ± 3 ^bc^	−2.0 ± 0.6 ^de^	10.9 ± 0.4 ^b^	11.1 ± 0.3 ^ab^	4.4 ± 2.4 ^a^
HAD60-30%	0.985 ± 0.003 ^a^	2.16 ± 0.03 ^a^	2.15 ± 0.03 ^a^	54 ± 3 ^a^	−1.3 ± 0.8 ^e^	11.23 ± 1.03 ^b^	11.5 ± 0.9 ^b^	4.5 ± 3.0 ^ab^
LYO-5%	0.989 ± 0.006	2.25 ± 0.04 ^ab^	2.21 ± 0.02 ^ab^	60.1 ± 0.9	−3.4 ± 0.2 ^cd^	10.8 ± 0.2 ^b^	10 ± 0.2 ^a^	2 ± 4 ^a^
LYO-10%	0.991 ± 0.002 ^bc^	2.261 ± 0.011 ^b^	2.259 ± 0.003 ^bc^	58.6 ± 0.8 ^bc^	−2.6 ± 0.3 ^de^	11.2 ± 0.5 ^b^	11.5 ± 0.5 ^b^	2.7 ± 2.7 ^a^
LYO-15%	0.987 ± 0.002 ^ab^	2.51 ± 0.02 ^d^	2.518 ± 0.006 ^ab^	59 ± 3 ^bc^	−6.7 ± 0.3 ^b^	14.0 ± 0.5 ^c^	15.4 ± 0.5 ^c^	3 ± 2 ^a^
LYO-20%	0.987 ± 0.002 ^ab^	2.41 ± 0.08 ^c^	2.45 ± 0.09 ^ab^	60.77 ± 0.96 ^c^	−6.7 ± 0.4 ^c^	15.0 ± 0.4 ^d^	16.3 ± 0.2 ^cd^	4± 3 ^ab^
LYO-25%	0.989 ± 0.003 ^ab^	2.521 ± 0.003 ^d^	2.53 ± 0.03 ^abc^	60.4 ± 1.2 ^c^	−5.6 ± 0.7 ^b^	16.5 ± 0.8 ^g^	17.4 ± 0.6 ^de^	4 ± 3 ^ab^
LYO-30%	0.988 ± 0.003 ^ab^	2.65 ± 0.04 ^ef^	2.62 ± 0.06 ^ab^	60.0 ± 2.4 ^c^	−6.1 ± 0.5 ^b^	15.8 ± 0.5 ^ef^	16.9 ± 0.6 ^d^	4.2 ± 2 ^ab^

The results represent the mean of three repetitions. Different lowercase letters within each storage condition (fresh or 7-day stored bread) indicate significant differences (*p* < 0.05). aw: water activity; relative weight was calculated as the ratio of the weight of bread prepared with the flour blend compared to the weight of bread prepared with rice flour (fresh or stored, as appropriate); L*a*b*C*: colour parameters; ∆E: colour differences.

**Table 3 foods-14-02382-t003:** Changes in functional properties of fresh and 7-day stored bread made from flour blends in which rice flour was partially replaced (5–30%) with almond bagasse powder, hot air-dried at 60 °C (HAD60) or lyophilised (LYO).

	Fresh Bread		7-day Stored Bread	
	HAD60	LYO	HAD60	LYO
Antioxidant activity (DPPH)	15–738% (**386%**)	188–580% (**402%**)	55–1805% (**907%**)	47–1649% (**877%**)
Antioxidant activity (FRAP)	137–621% (**307%**)	180–615% (**315%**)	88–570% (**297%**)	103–521% (**287%**)
Total phenol content	40–134% (**82%**)	20–147% (**86%**)	39–157% (**95%**)	25–142% (**84%**)
Reducing sugars	88–134% (**108%**)	38–72% (**53%**)	143–202% (**168%**)	25–117% (**77%**)

The values represent the minimum–maximum range of increase relative to the control (bread made with 100% rice flour), and the values in parentheses indicate the average across all substitution levels (5, 10, 15, 20, 25, and 30%).

## Data Availability

The original contributions presented in this study are included in the article; further inquiries can be directed to the corresponding author.

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
