# Peer review of "Enrichment of Rice Flour with Almond Bagasse Powder: The Impact on the Physicochemical and Functional Properties of Gluten-Free Bread"

_foods, 2025, doi:10.3390/foods14132382_

Round 1
Reviewer 1 Report
Comments and Suggestions for Authors
This manuscript presents a comprehensive and valuable study on the utilization of almond bagasse powder, a significant by-product, as a functional ingredient in gluten-free bread. The authors have conducted an extensive set of experiments, covering the physicochemical, techno-functional, rheological, and antioxidant properties of both the flour blends and the final bread products (fresh and stored). The comparison between two different drying methods (hot air-drying vs. lyophilisation) is a notable strength, providing deeper insights into how processing affects the functional attributes of the ingredient. The manuscript is generally well-structured, and the data presented are extensive. However, despite the study's strengths, there are several significant points that require clarification and revision before the manuscript can be considered for publication.
- Introduction, Lines 80-82. The text states that incorporating almond bagasse "has shown improvements in both nutritional and sensory attributes." To strengthen the introduction's argument, it would be beneficial to be slightly more specific. For example, what kind of sensory improvements were noted in the cited study? (e.g., improved crunchiness, flavor, etc.). This is a minor point but would add valuable context.
- Section 2.2. Bread production using almond bagasse powder, Lines 117-121. The manuscript states that the control rice flour bread used water corresponding to 44% hydration. However, it is not specified how the water content was determined for the formulations containing almond bagasse powder. The results in Table 1 clearly show that the Water Absorption Capacity (WAC) of the flour blends changes significantly with the substitution level. Did the authors use a constant amount of water for all formulations, or was the water addition adjusted based on the WAC of each specific flour blend? This is a critical detail. If a constant water level was used, the effective hydration of the doughs would differ, confounding the results for specific volume, dough rheology, and bread texture. If the water was adjusted, this methodology (e.g., based on Farinograph or empirical tests) must be explicitly described. This lack of information is a major flaw that undermines the comparability of the different bread formulations.
- Section 3.1, Lines 286-288 and Table 1. The text states, "for OAC, no statistically significant differences were observed." This is immediately followed by, "However, a downward trend in OAC was observed when almond bagasse powder was incorporated..." These statements are somewhat contradictory. More importantly, the statistical lettering in Table 1 does show significant differences among some of the samples (e.g., Rice flour 1.57ab vs. LYO-5% 1.73d). The discussion should be revised to accurately reflect the statistical analysis presented in the table. The authors should precisely describe which formulations differed significantly from the control and from each other, rather than making a blanket statement.
- Section 3.1, Lines 346-351 and Table 2 footnote. The footnote in Table 2 defines relative weight as "the ratio of the weight of bread prepared with the flour blend compared to the weight of bread prepared with rice flour." For fresh bread, this is clear. However, for 7-day stored bread, the text states that values were "greater than 1, suggesting that the substituted breads weighed more than the control." Is the denominator in this calculation the weight of the fresh control bread or the stored control bread? This must be clarified. If it is the stored control, the interpretation that substituted breads retained more moisture is valid. If it is the fresh control, the interpretation is less direct. Please clarify the calculation basis for the stored bread's relative weight.
- Section 3.4, comparison between Figure 5d (Lines 613-623) and Figure 6d (Lines 700-711). There is a clear contradiction that needs to be addressed. In the analysis of the flours (Figure 5d), the text correctly states that LYO formulations had a higher content of reducing sugars than HAD60 formulations. However, in the analysis of the fresh bread (Figure 6d), the text states the opposite: "breads incorporating hot air-dried powder (HAD60) showed higher levels of reducing sugars than those prepared with lyophilised powder (LYO)." The authors must explain this reversal. A plausible hypothesis is that the higher initial sugar content in LYO flours was more readily consumed during Maillard/caramelization reactions during baking, while the HAD60 powder, having already undergone some thermal treatment, might release sugars more slowly or differently during baking. This needs to be explicitly discussed and hypothesized upon, as it is a very interesting but currently unexplained finding.
- Section 3.3, Lines 545-548. The authors astutely observe that for stored breads, "the second compression was performed on an already fractured bread sample, which affected the force values recorded." This is an important limitation of the TPA method for highly brittle samples. The authors should briefly comment on how this might affect the interpretation of the second compression parameters (like cohesiveness, which is derived from it) for the most brittle samples (e.g., stored control). Does this make the second compression data for these specific samples less reliable or not directly comparable to the less brittle samples? Acknowledging this limitation would add rigor to the analysis.
- Section 3.2, Lines 386-388 and Figure 1. The text makes a general statement that "the incorporation of almond bagasse powder... affects the viscosity profile regardless of the percentage of substitution." While broadly true, the ANOVA results provided in the nested table of Figure 1 show that the effect of treatment type (A: HAD60 vs. LYO) is statistically significant for Pasting Temperature (p=0.0412). The discussion should be more nuanced to reflect the statistical findings. For instance, the authors could mention that while substitution level was the dominant factor for most viscosity parameters, the drying method also had a minor but significant effect on the gelatinization temperature. In addition, the nested tables in Figure 1 should be presented as separate tables.
Author Response
|
RESPONSE TO REVIEWERS Thank you very much for taking the time to review this manuscript. Please find the detailed responses below and the corresponding revisions/corrections in track changes in the re-submitted files |
REVIEWER 1 COMMENTS
This manuscript presents a comprehensive and valuable study on the utilization of almond bagasse powder, a significant by-product, as a functional ingredient in gluten-free bread. The authors have conducted an extensive set of experiments, covering the physicochemical, techno-functional, rheological, and antioxidant properties of both the flour blends and the final bread products (fresh and stored). The comparison between two different drying methods (hot air-drying vs. lyophilisation) is a notable strength, providing deeper insights into how processing affects the functional attributes of the ingredient. The manuscript is generally well-structured, and the data presented are extensive. However, despite the study's strengths, there are several significant points that require clarification and revision before the manuscript can be considered for publication.
- Introduction, Lines 80-82. The text states that incorporating almond bagasse "has shown improvements in both nutritional and sensory attributes." To strengthen the introduction's argument, it would be beneficial to be slightly more specific. For example, what kind of sensory improvements were noted in the cited study? (e.g., improved crunchiness, flavor, etc.). This is a minor point but would add valuable context.
Thank you for your appreciation.
The authors of the research detected effects on sensory properties at the pilot test level. However, as the results are not part of a structured, scientifically valid study, the statement has been removed from the text.
- Section 2.2. Bread production using almond bagasse powder, Lines 117-121. The manuscript states that the control rice flour bread used water corresponding to 44% hydration. However, it is not specified how the water content was determined for the formulations containing almond bagasse powder. The results in Table 1 clearly show that the Water Absorption Capacity (WAC) of the flour blends changes significantly with the substitution level. Did the authors use a constant amount of water for all formulations, or was the water addition adjusted based on the WAC of each specific flour blend? This is a critical detail. If a constant water level was used, the effective hydration of the doughs would differ, confounding the results for specific volume, dough rheology, and bread texture. If the water was adjusted, this methodology (e.g., based on Farinograph or empirical tests) must be explicitly described. This lack of information is a major flaw that undermines the comparability of the different bread formulations.
Thank you for the observation. While Table 1 shows that the Water Absorption Capacity (WAC) of flour blends varies with substitution level, maintaining constant hydration allowed for direct comparison of how almond bagasse affects dough rheology, bread structure, and quality parameters independent of water optimization effects. We added the following description in 2.2. subchapter (lines 137-147).“Subsequently, breads containing almond bagasse powder, either air-dried or lyophilised, were prepared as partial substitutes for the rice flour. In these formulations, a constant water content of 44% hydration was maintained across all treatments (consistent with the control formulation), with 100 g of flour mixture as the base, and substitution levels of 5, 10, 15, 20, 25, and 30% (w/w). This approach was deliberately chosen to evaluate the specific impact of almond bagasse substitution on bread matrix properties under standardized hydration conditions, rather than optimizing water content for each individual blend.”
- Section 3.1, Lines 286-288 and Table 1. The text states, "for OAC, no statistically significant differences were observed." This is immediately followed by, "However, a downward trend in OAC was observed when almond bagasse powder was incorporated..." These statements are somewhat contradictory. More importantly, the statistical lettering in Table 1 does show significant differeªnces among some of the samples (e.g., Rice flour 1.57ab vs. LYO-5% 1.73d). The discussion should be revised to accurately reflect the statistical analysis presented in the table. The authors should precisely describe which formulations differed significantly from the control and from each other, rather than making a blanket statement.
OK, a more accurate discussion has been included in lines 327-338.
- Section 3.1, Lines 346-351 and Table 2 footnote. The footnote in Table 2 defines relative weight as "the ratio of the weight of bread prepared with the flour blend compared to the weight of bread prepared with rice flour." For fresh bread, this is clear. However, for 7-day stored bread, the text states that values were "greater than 1, suggesting that the substituted breads weighed more than the control." Is the denominator in this calculation the weight of the fresh control bread or the stored control bread? This must be clarified. If it is the stored control, the interpretation that substituted breads retained more moisture is valid. If it is the fresh control, the interpretation is less direct. Please clarify the calculation basis for the stored bread's relative weight.
For 7-day stored bread, the denominator in the calculation of relative weight was the weight of the stored control bread. To clarify, the authors have completed the footnote and materials and methods section with the statement “fresh or stored, as appropriate”.
- Section 3.4, comparison between Figure 5d (Lines 613-623) and Figure 6d (Lines 700-711). There is a clear contradiction that needs to be addressed. In the analysis of the flours (Figure 5d), the text correctly states that LYO formulations had a higher content of reducing sugars than HAD60 formulations. However, in the analysis of the fresh bread (Figure 6d), the text states the opposite: "breads incorporating hot air-dried powder (HAD60) showed higher levels of reducing sugars than those prepared with lyophilised powder (LYO)." The authors must explain this reversal. A plausible hypothesis is that the higher initial sugar content in LYO flours was more readily consumed during Maillard/caramelization reactions during baking, while the HAD60 powder, having already undergone some thermal treatment, might release sugars more slowly or differently during baking. This needs to be explicitly discussed and hypothesized upon, as it is a very interesting but currently unexplained finding.
Ok, thank you very much for your input. The authors agree with the justification provided by the reviewer and have incorporated it into the discussion (lines 827-835).
- Section 3.3, Lines 545-548. The authors astutely observe that for stored breads, "the second compression was performed on an already fractured bread sample, which affected the force values recorded." This is an important limitation of the TPA method for highly brittle samples. The authors should briefly comment on how this might affect the interpretation of the second compression parameters (like cohesiveness, which is derived from it) for the most brittle samples (e.g., stored control). Does this make the second compression data for these specific samples less reliable or not directly comparable to the less brittle samples? Acknowledging this limitation would add rigor to the analysis.
We appreciate this observation regarding the limitations of TPA analysis for highly brittle samples. You are absolutely correct that fracturing during the first compression affects the reliability and interpretation of second compression-derived parameters. We have added a discussion of this limitation and its implications for data interpretation in the revised manuscript (lines 647-657).
- Section 3.2, Lines 386-388 and Figure 1. The text makes a general statement that "the incorporation of almond bagasse powder... affects the viscosity profile regardless of the percentage of substitution." While broadly true, the ANOVA results provided in the nested table of Figure 1 show that the effect of treatment type (A: HAD60 vs. LYO) is statistically significant for Pasting Temperature (p=0.0412). The discussion should be more nuanced to reflect the statistical findings. For instance, the authors could mention that while substitution level was the dominant factor for most viscosity parameters, the drying method also had a minor but significant effect on the gelatinization temperature. In addition, the nested tables in Figure 1 should be presented as separate tables.
Ok, the statistical specification has now been incorporated in the text. Regarding the nested tables in Figure 1, we believe they should not be separated, as they complement the graph and stem directly from it. Therefore, it is preferable to keep them together to preserve clarity and coherence.

Reviewer 2 Report
Comments and Suggestions for Authors
Manuscript review
- Could the authors provide a quantitative estimation of almond bagasse availability as a by-product in the food industry, to contextualize the applicability potential of this study?
- The introduction states that incorporating up to 5% of by-products does not significantly alter flavor or texture. However, the study presents formulations ranging from 5% to 30%. Could the authors justify the selection of such high substitution levels with almond bagasse?
- Could the authors include a table showing the nutritional profile of almond bagasse used in the study, highlighting its fiber, fat, protein and other relevant contents? This would help assess its potential as a functional ingredient.
- Although the advantages of using almond bagasse are highlighted in the introduction, it would be helpful to also mention some possible drawbacks, such as:
- It may impart a slightly bitter or residual taste if not properly processed.
- Its high fat content makes it prone to oxidation and rancidity.
- There may be variability in composition depending on source and processing method.
- It requires additional treatments (drying, grinding) that increase costs and may affect functional properties.
- Line 129: Could the authors specify more precisely the exact refrigeration temperature used during sample storage?
- Line 135: Could the authors correctly cite the standard ICC 162 method in the references, according to AACC International (2000), 11th Edition?
- Reducing Sugars: Could the authors indicate in the experimental section the source of the reagents used and the specific model of the spectrophotometer employed in the analysis?
- Pasting: Could the authors include the ICC 162 method in the list of bibliographic references, since it is mentioned as the methodological basis for viscosity analysis?
- Could the authors comment on whether there are strategies to mitigate the observed color changes, such as modifying processing parameters, adding antioxidants, or adjusting pH? Color is a critical parameter for sensory acceptance in flour-based products.
- Why was pH not evaluated in the blends? Since almond bagasse could influence color changes, microbial growth, or the Maillard reaction, including pH in the analysis would be relevant. If data is available, could it be added to the manuscript?
- In the pasting properties analysis, peak viscosity (PV), trough viscosity (TV), final viscosity (FV), and setback viscosity (SV) were found to be lower in samples with lyophilized almond bagasse (LYO) compared to HAD60. Could the authors provide a more detailed explanation for this difference?
- A relationship between pasting properties and the fiber content of almond bagasse is observed. Could the authors summarize these trends in an integrated way to facilitate understanding of how fiber influences those properties?
- While the addition of almond bagasse improves the nutritional profile (increased fiber), it also reduces firmness and elasticity of gels and breads, especially at high concentrations. What strategies do the authors propose to minimize this negative impact on texture? For example, could the use of additives such as gums, combination with other flours, or finer milling be viable solutions?
- Would the authors recommend combining lyophilized and hot air-dried almond bagasse (HAD60) to balance the positive and negative effects on the final product characteristics?
- Could the authors include a comparative table summarizing the behavior of the following parameters according to the almond bagasse production method (HAD60 vs LYO)? The table should include increases, decreases, and possible causes to facilitate interpretation of the results:
- Antioxidant activity (DPPH)
- Antioxidant activity (FRAP)
- Total phenolic content
- Reducing sugars
- Antioxidant capacity (DPPH and FRAP)
- Formation of new antioxidants
- Preservation of natural phenols
- Release during baking
- After reading the conclusion, do the authors consider that a mixture of almond bagasse obtained by both methods could yield better results? In addition, could they suggest in the conclusion possible modifications or future research lines aimed at optimizing the incorporation of almond bagasse into rice flour?
Author Response
|
Thank you very much for taking the time to review this manuscript. Please find the detailed responses below and the corresponding revisions/corrections in track changes in the re-submitted files |
REVIEWER 2 COMMENTS
Manuscript review
- Could the authors provide a quantitative estimation of almond bagasse availability as a by-product in the food industry, to contextualize the applicability potential of this study?
The authors could not find any direct information on the volume of by-product generated. However, its generation is associated with the production of the almond vegetable drink, and there are production and growth data for the coming years for this drink. The following paragraph has been added to the introduction (lines 48-52):
“It is a by-product whose generation is associated with the production of the almond vegetable drink, and consequently its increase is associated with the increase in its production. It is worth noting that the almond milk market was estimated at $5.49 bil-lion in 2024 and is expected to reach $9.61 billion by 2029, growing at a CAGR of 11.85% over the period from 2024 to 2029 (Mordor Intelligence, 2025).”
Source: https://www.mordorintelligence.com/es/industry-reports/global-almond-milk-market
- The introduction states that incorporating up to 5% of by-products does not significantly alter flavor or texture. However, the study presents formulations ranging from 5% to 30%. Could the authors justify the selection of such high substitution levels with almond bagasse?
The benefits of valorising the by-product and improving the nutritional value of gluten-free bread would increase with a higher level of substitution. Additionally, while high substitution levels may alter the sensory properties of the final product, almonds are a nut whose sensory properties are valued by consumers, suggesting that the changes could be favourable. The justification has been added to the text (lines 87-99).
- Could the authors include a table showing the nutritional profile of almond bagasse used in the study, highlighting its fiber, fat, protein and other relevant contents? This would help assess its potential as a functional ingredient.
The same authors published the information in open access in the following reference:
Duarte, S.; Betoret, E.; Barrera, C.; Seguí, L.; Betoret, N. Integral valorization of almond bagasse by dehydration: Physicochemical and technological properties and modeling of hot air drying. Sustainability 2023, 15, 10704. https://doi.org/10.3390/su151310704
Reference has been incorporated into the text.
- Although the advantages of using almond bagasse are highlighted in the introduction, it would be helpful to also mention some possible drawbacks, such as:
- It may impart a slightly bitter or residual taste if not properly processed.
- Its high fat content makes it prone to oxidation and rancidity.
- There may be variability in composition depending on source and processing method.
- It requires additional treatments (drying, grinding) that increase costs and may affect functional properties.
Of course. The next paragraph has been added to the introduction (lines 90-99).
“However, incorporating this ingredient can also lead to the final product having undesirable characteristics. Its high fat and fibre content, together with the formation of compounds during air drying, could impart residual and rancid flavours to the final product. Furthermore, It requires additional treatments that increase production costs.”
- Line 129: Could the authors specify more precisely the exact refrigeration temperature used during sample storage?
Yes, the temperature was 4 ºC and the data has been added to the text (Line 159).
- Line 135: Could the authors correctly cite the standard ICC 162 method in the references, according to AACC International (2000), 11th Edition?
Certainly, the citation has now been properly included in the manuscript in accordance with your suggestion.
- Reducing Sugars: Could the authors indicate in the experimental section the source of the reagents used and the specific model of the spectrophotometer employed in the analysis?
Thank you for your observation. The source of the reagents has been included, and the spectrophotometer model has been specified.
- Pasting: Could the authors include the ICC 162 method in the list of bibliographic references, since it is mentioned as the methodological basis for viscosity analysis?
Of course, the citation has been duly included in the manuscript.
- Could the authors comment on whether there are strategies to mitigate the observed color changes, such as modifying processing parameters, adding antioxidants, or adjusting pH? As the color changes are associated to browning reactions, it can be avoided by modifying air drying temperature. This effect was studied by the authors in another work showing that HAD at 70 ºC was the most appropriated temperature to stabilize raw material while colour differences were minimum. The reference is:
Duarte, S.; Betoret, E.; Barrera, C.; Seguí, L.; Betoret, N. Integral Recovery of Almond Bagasse through Dehydration: Physico-Chemical and Technological Properties and Hot Air-Drying Modelling. Sustainability 2023, 15, 10704. https://doi.org/10.3390/su151310704
- Why was pH not evaluated in the blends? Since almond bagasse could influence color changes, microbial growth, or the Maillard reaction, including pH in the analysis would be relevant. If data is available, could it be added to the manuscript?
The authors decided not to consider the pH of almond bagasse because it has a pH similar to that of rice flour and therefore will not be affected by the substitution.
- In the pasting properties analysis, peak viscosity (PV), trough viscosity (TV), final viscosity (FV), and setback viscosity (SV) were found to be lower in samples with lyophilized almond bagasse (LYO) compared to HAD60. Could the authors provide a more detailed explanation for this difference?
We agree that an integrated summary of fiber-pasting property relationships would greatly enhance understanding. We have added a a discussion (lines 468-484).
- A relationship between pasting properties and the fiber content of almond bagasse is observed. Could the authors summarize these trends in an integrated way to facilitate understanding of how fiber influences those properties?
We appreciate this important consideration, and we have added a discussion (lines 503-522).
- While the addition of almond bagasse improves the nutritional profile (increased fiber), it also reduces firmness and elasticity of gels and breads, especially at high concentrations. What strategies do the authors propose to minimize this negative impact on texture? For example, could the use of additives such as gums, combination with other flours, or finer milling be viable solutions?
Thank you, You are absolutely right that addressing the texture limitations while maintaining nutritional benefits is crucial for commercial viability. We have added a discusión in the section 3.5.
- Would the authors recommend combining lyophilized and hot air-dried almond bagasse (HAD60) to balance the positive and negative effects on the final product characteristics?
Thank you. The complementary characteristics of LYO and HAD60 powders indeed suggest that strategic combinations could balance their respective advantages and limitations. However, lyophilised samples are more expensive than hot-dried, which makes it economically questionable.
- Could the authors include a comparative table summarizing the behavior of the following parameters according to the almond bagasse production method (HAD60 vs LYO)? The table should include increases, decreases, and possible causes to facilitate interpretation of the results:
Thank you for the suggestion. A summary table has been included at the end of 3.4 section, highlighting the increases in antioxidant activity, total phenolic content, and reducing sugars according to the production method. The possible causes are emphasized in the table explanation.
Antioxidant activity (DPPH)
- Antioxidant activity (FRAP)
- Total phenolic content
- Reducing sugars
- Antioxidant capacity (DPPH and FRAP)
- Formation of new antioxidants
- Preservation of natural phenols
- Release during baking
- After reading the conclusion, do the authors consider that a mixture of almond bagasse obtained by both methods could yield better results? In addition, could they suggest in the conclusion possible modifications or future research lines aimed at optimizing the incorporation of almond bagasse into rice flour?
Yes, we absolutely agree that combining both drying methods could yield superior results by leveraging their complementary strengths. We have significantly expanded our conclusions.

Round 2
Reviewer 1 Report
Comments and Suggestions for Authors
Acceptance.
Reviewer 2 Report
Comments and Suggestions for Authors
The authors addressed the reviews appropriately. I am very pleased with their contributions. I send my congratulations.